# Polymer-Supported Oxidovanadium(IV) Complexes and Their Catalytic Applications in One-Pot Multicomponent Reactions Producing Biologically Active 2,4,5-Trisubstituted-1*H*-imidazoles

**Mannar R. Maurya** [1,*], **Monojit Nandi** [1], **Akhil Patter** [1], **Fernando Avecilla** [2] **and Kaushik Ghosh** [1]

[1] Department of Chemistry, Indian Institute of Technology Roorkee, Roorkee 247667, India
[2] Nano ToxGen Group, Department of Chemistry, Interdisciplinary Centre for Chemistry and Biology (CICA), Science Faculty, Campus Zapateira, University of Coruña, 15071 A Coruña, Spain
* Correspondence: m.maurya@cy.iitr.ac.in

**Abstract:** Two new monobasic tridentate O⌒N⌒N donor ligands, HL$_1$ (**I**) and HL$_2$ (**II**) have been obtained in two steps by reacting phenylhydrazine and salicylaldehyde or 3,5-di-*tert*-butylsalicylaldehyde and then reacting the resulting compounds with 2-chloromethylbenzimidazole in the presence of triethylamine. The reaction of [V$^{IV}$O(acac)$_2$] with these ligands in a 1:1 molar ratio in dry methanol led to the formation of homogeneous oxidovanadium(IV) complexes [V$^{IV}$O(acac)L$_1$] (**1**) and [V$^{IV}$O(acac)L$_2$] (**2**). Immobilization of these complexes on chloromethylated polystyrene (PS-Cl) cross-linked with divinyl benzene resulted in corresponding polymer-supported heterogeneous complexes PS-[V$^{IV}$O(acac)L$_1$] (**3**) and PS-[V$^{IV}$O(acac)L$_2$] (**4**). Ligands (**I** and **II**), homogeneous complexes (**1** and **2**) and heterogeneous complexes (**3** and **4**) have been characterized using elemental analysis and various spectroscopic techniques. A single crystal X-ray diffraction study of **I** and **1** further confirms their structures. The oxidation state IV of vanadium in these complexes was assured by recording their EPR spectra while heterogeneous complexes were further characterized using field emission-scanning electron microscopy (FE-SEM) combined with energy dispersive X-ray analysis (EDS) and atomic force microscopy (AFM). All vanadium complexes have been explored for their catalytic potential to one-pot-three-component reactions (reagents: benzil, ammonium acetate and various aromatic aldehydes) for the efficient synthesis of 2,4,5-triphenyl-1*H*-imidazole derivatives (nine examples). Various reaction conditions have been optimized to obtain a maximum yield (up to 96%) of catalytic products. It has been found that heterogeneous complexes show excellent catalytic activity and are recyclable up to five catalytic cycles.

**Keywords:** heterogeneous catalysts; vanadium complexes; one pot multicomponent reaction; single crystal X-ray study; spectroscopy





## 1. Introduction

In the past few decades, nitrogen-containing, five-membered heterocyclic compounds having imidazole moiety, specifically 2,4,5-trisubstituted imidazole molecules, have received enormous interest from researchers due to their unique chemical behaviors and tremendous applications in pharmaceutical industries [1–5]. In nature, several biologically active heterocyclic compounds contain imidazole functional groups in their core structure such as biotin, histidine, histamine, peptides, alkaloids and nucleic acids that have important pharmacological properties [6–8]. In medicinal chemistry, imidazole-based five-membered heterocyclic compounds with an imidazole core structure show a broad range of therapeutic applications including antifungal [9,10], antibacterial [11,12], anticancer [13–15], anti-inflammatory [16,17], anthelmintic and cytotoxic activity [18], anti-neurodegenerative and neuroprotective agent [19,20], antiviral [21], antitumor [22], cholesterol-lowering

agent [23,24], antiulcer [25,26], antidiabetic [27,28], local anesthetic agents [29], antiobesity [30], antihyperuricemic [31], antileishmanial [32], antimicrobial [33,34], antiproliferative [35–37], inhibitors of P38 MAP kinase [38] and antihypertensive [39]. Imidazole containing some important molecules having therapeutic applications are shown in Figure 1 [24,40–43]. In addition, in the recent past, imidazole derivatives have also received great attention from researchers due to their use in organometallic chemistry, e.g., synthesis of *N*-heterocyclic carbenes [44,45], and in green chemistry such as a green solvent [46,47].

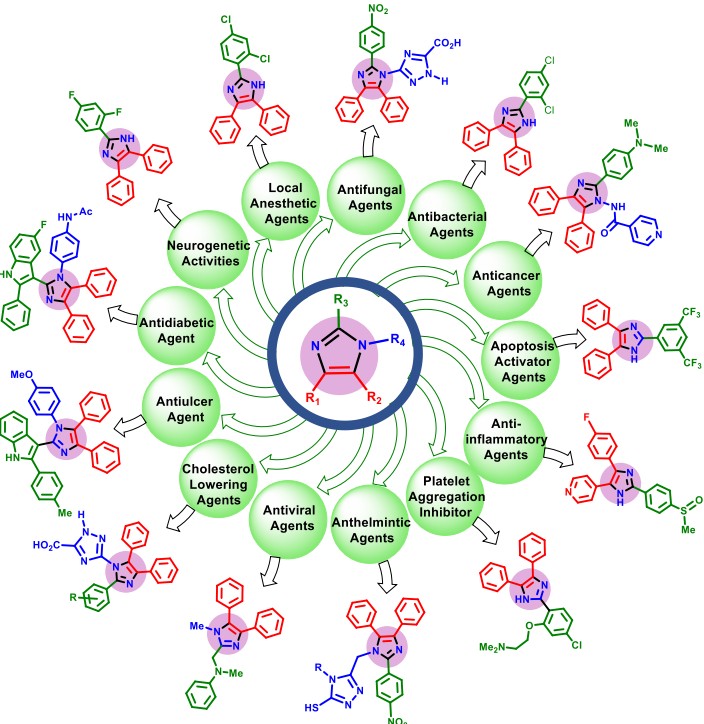

**Figure 1.** Schematic presentation of some biologically important 2,4,5-trisubstituted imidazole containing heterocyclic compounds having therapeutic applications.

Nowadays, instead of using conventional catalysts such as Lewis or Brønsted acid and bases [48–51], the new generation of catalysts such as supported catalysts [52], polymer-based catalysts [53], metal–organic frameworks [54] and enzymes [55] have been used for various organic and inorganic transformations. In the literature, many synthetic methods have been reported for one-pot-multicomponent reactions of benzil, ammonium acetate and different aromatic aldehyde to synthesize 2,4,5-trisubstituted imidazoles under different reaction conditions [56–72]. In general, most procedures in the literature have their own advantages, but there are still some drawbacks in the literature procedures such as the use of non-reusable catalysts, long reaction times, hazardous solvents, harsh reaction conditions, high temperatures, the requirement of a large amount of catalyst and low yield of products.

For the development of new efficient catalytic methods to overcome these drawbacks and to provide more atom–economic, environmental-friendly and industrially favorable reaction conditions, we have designed polymer-supported heterogeneous vanadium complex-based catalysts (see Sections 2.1 and 2.2). These catalysts have been used for one-pot multicomponent reactions (MCRs) (reagents: benzil, ammonium acetate and various aromatic aldehydes) for the synthesis of 2,4,5-trisubstituted imidazole molecules in a non-hazardous solvent (EtOH). The catalysts designed herein produce a high yield (up to 96%) in less reaction time (ca. 30 min) and are reusable for up to five cycles. Analogous homogeneous oxidovanadium(IV) complexes have also been prepared, characterized and used for the above catalytic reaction.

## 2. Results and Discussion

### 2.1. Synthesis and Structural Characterization of Ligands

Ligands $HL_1$ (**I**) and $HL_2$ (**II**) are isolable in two steps using the reaction of phenylhydrazine with salicylaldehyde or 3,5-di-*tert*-butylsalicylaldehyde followed by reacting the resulting compounds with 2-chloromethylbenzimidazole in the presence of triethylamine in dry MeOH (Scheme 1).

**Scheme 1.** Synthetic scheme for the preparation of tridentate monobasic N⌢N⌢O donor ligands **I** and **II**.

The elemental analysis, spectroscopic characterization (IR, UV-visible and NMR) and single-crystal X-ray diffraction study for **I** confirmed (vide infra) the structure of ligands.

The Supporting Materials section contains characterization details of the ligands. The protons of N–H and O–H appeared as a singlet at 12.38 and 10.46 ppm, respectively, for **I** and at 12.46 and 11.84 ppm, respectively, for **II**. Peaks at 5.43 ppm in **I** and 5.54 ppm in **II** clearly indicated the presence of aliphatic protons ($-CH_2$ group). The proton of the imine group (–HC=N–) appears as a singlet at 8.00 and 8.19 ppm, respectively, in these ligands. Due to benzene rings, protons appear in the usual range of 6.80–7.53 ppm (for **I**) and 6.99–7.40 ppm (for **II**) expected for aromatic compounds. The $^{13}$C-NMR spectra of these ligands exhibit signals due to aromatic carbons between 111.94 and 156.09 ppm, aliphatic carbons at 45.09 ppm (for **I**) and 44.54 (for **II**) ppm and carbon due to the azomethine group at 150.36 ppm (for **I**) and 143.79 ppm (for **II**). The tert-butyl groups of **II** appear between 29.91 and 35.15 ppm. All these confirm the structure of the ligands as proposed.

### 2.2. Synthesis and Solid-State Characterization of Complexes

The reaction between $[V^{IV}O(acac)_2]$ and ligands **I** and **II** in refluxing dry MeOH led to the formation of oxidovanadium(IV) complexes $[V^{IV}O(acac)L_1]$ (**1**) and $[V^{IV}O(acac)L_2]$ (**2**), respectively. These complexes have further been immobilized on chloromethylated polystyrene cross-linked with divinylbenzene (PS–Cl) by reacting them with polystyrene in DMF in a slightly basic medium. Here, the chloro group of PS–Cl reacted with the de-protonatable proton of the imidazole group present on the ligand part of the complexes to give PS-$[V^{IV}O(acac)L_1]$ (**3**) and PS-$[V^{IV}O(acac)L_2]$ (**4**). Complete synthetic procedures and proposed structures of homogeneous as well as heterogeneous complexes are given in Scheme 2. The structures of the homogeneous complexes are based on elemental and thermal analyses, a spectroscopic (IR, UV-visible and EPR) study and a single crystal X-ray diffraction study of **1**. The structures of the heterogeneous complexes were additionally characterized using microwave plasma atomic emission spectroscopy (MP-AES), field emission-scanning electron microscopy (FE-SEM), energy-dispersive spectroscopy (EDS) and atomic force microscopy (AFM) analyses.

**Scheme 2.** Scheme for the synthesis of homogeneous and heterogeneous oxidovanadium(IV) complexes reported in this work.

## 2.3. Thermal Study

The TGA profiles of all four complexes are shown in Figure 2. Both homogeneous complexes are stable up to 230 °C and thereafter, they lose 19.6% (**1**) and 15.9% (**2**) mass within the next 335–340 °C temperature range in two overlapping steps due to the rupturing of the acetylacetone group, as the calculated value of 19.5% and 16.0%, respectively, matches well with the observed one. Immediately, the decomposition of the ligand starts upon further increasing the temperature and continues until 527 °C for **1** and 532 °C for **2**. The obtained residue of 17.7% in **1** (calc. 17.9%) and 14.7% in **2** (calc. 14.7%) is close to the calculated value for $V_2O_5$.

The heterogeneous complexes **3** and **4** are stable up to 170 °C and thereafter, both complexes decompose in three overlapping steps. The initial mass loss of approximately 2.7 and 3.3% is possibly due to the adsorbed moisture. After 170 °C, a part of the ligand decomposition (16.2% mass loss at 258 °C in complex **3** and 13.2 % at 255 °C in complex **4**) was observed. This decomposition overlaps with the decomposition of the polymer backbone and completes at 370 °C in **3** and at 375 °C in **4**. No further mass loss was noted beyond these temperatures. After this temperature, the decomposition stabilized, and the final residue of 7.1% in **3** and 7.5% in **4** are due to $V_2O_5$. The vanadium content calculated from the $V_2O_5$ residue was found to be 0.78 mmol/g in **3** and 0.83 mmol/g in **4**, which are in good agreement with the results obtained using MP-AES.

## 2.4. Structure Description of $HL_1$ (**I**) and $[V^{IV}O(acac)L_1]$ (**1**)

The structure of ligand $HL_1$ (**I**) and its vanadium complex $[V^{IV}O(acac)L_1]$ (**1**) were characterized using single-crystal X-ray diffraction study. Selected bond lengths and angles are reported in Tables S1 and S2 (see the Supporting Information). The ORTEP representations of these compounds are shown in Figures 3 and 4. **I** crystallized in a non-centrosymmetric orthorhombic space group, *Fdd2*, while **1** crystallized in a centrosymmetric monoclinic space group, *P21/c*.

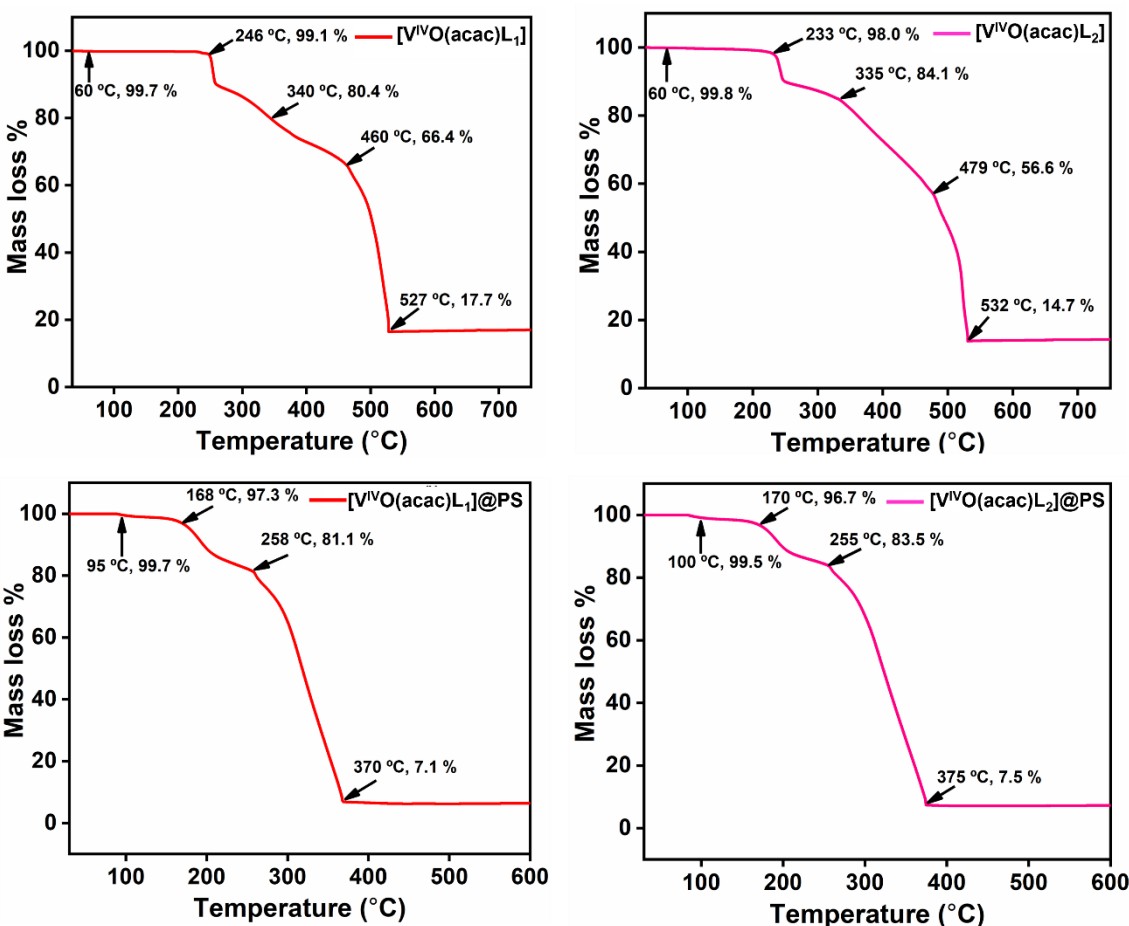

**Figure 2.** TGA profiles of complexes **1**, **2**, **3** and **4**.

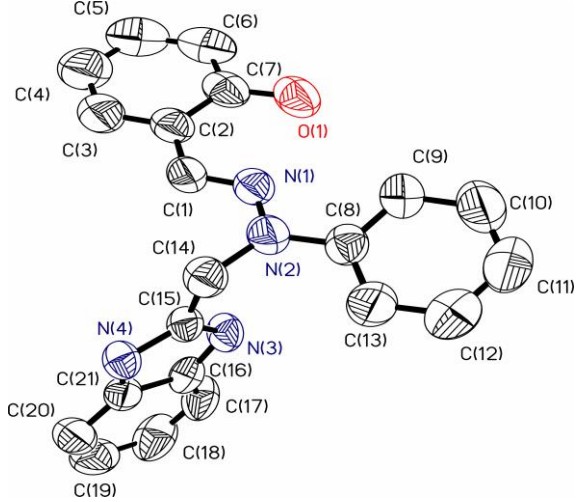

**Figure 3.** ORTEP of the asymmetric unit in ligand $HL_1$ (**I**). Non-hydrogen atoms were drawn using 50% probability ellipsoids. Hydrogen atoms were omitted for clarity. The drawing was made with the SHELXL set of programs. CCDC number: 2241272.

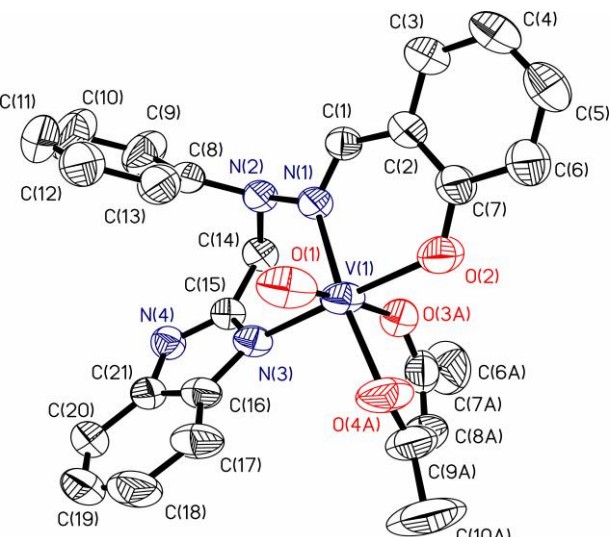

**Figure 4.** ORTEP of the asymmetric unit in **1**. Non-hydrogen atoms were drawn using 50% probability ellipsoids. Hydrogen atoms were omitted for clarity. The drawing was made with the SHELXL set of programs. CCDC number: 2241273.

The asymmetric unit of **I** contains only one molecule, but the asymmetric unit of **1** contains two vanadium complexes (see Figure 5). This may be because the two vanadium complexes, present in the asymmetric unit, are not exactly the same. In the ligand, two enantiomeric pairs were observed (see Figure 6), but in the case of **1**, the two enantiomers change their conformation slightly (see Figure 7). If we determine the dihedral angles between the rings of the ligand, the angle between the phenyl ring and benzimidazole ring is 87.58(5)°, the angle between the benzimidazole ring and phenolate ring is 87.80(5)° and the angle between the phenyl ring and phenolate ring is 15.88(9)°. In **1** (see Figure 5), the same angles in the vanadium complex (a) are 47.77(6)°, 54.52(7)° and 92.87(6)°, respectively, and in the vanadium complex (b) are 15.37(10)°, 78.02(5)° and 93.31(8)°, respectively.

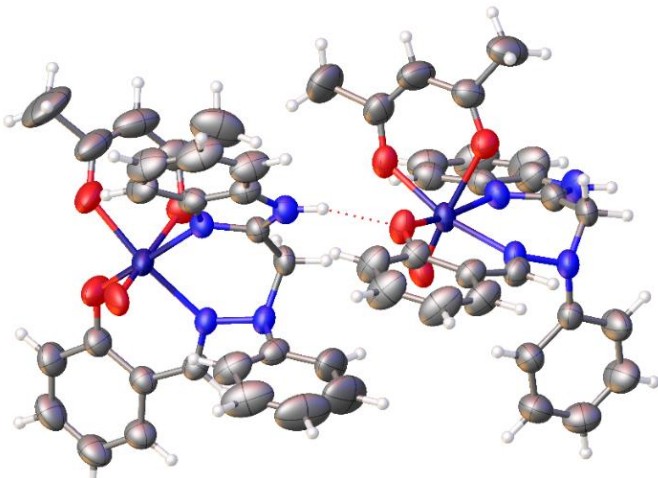

**Figure 5.** Asymmetric unit of compound **1** showing two molecules of the complex.

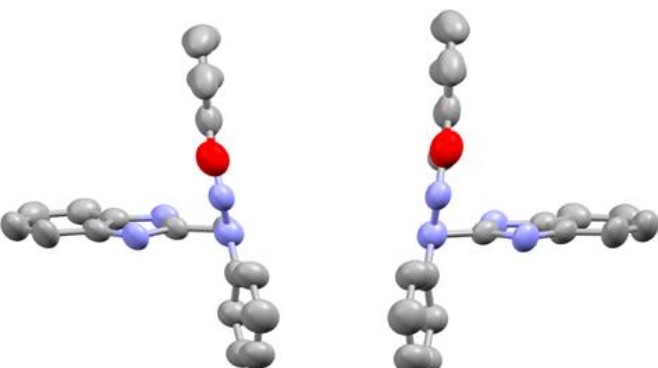

**Figure 6.** View of enantiomer pairs present in the crystal packing in HL$_1$. Non-hydrogen atoms were drawn using 50% probability ellipsoids. Hydrogen atoms were omitted for simplicity. The drawings were made with the mercury 3.8 program.

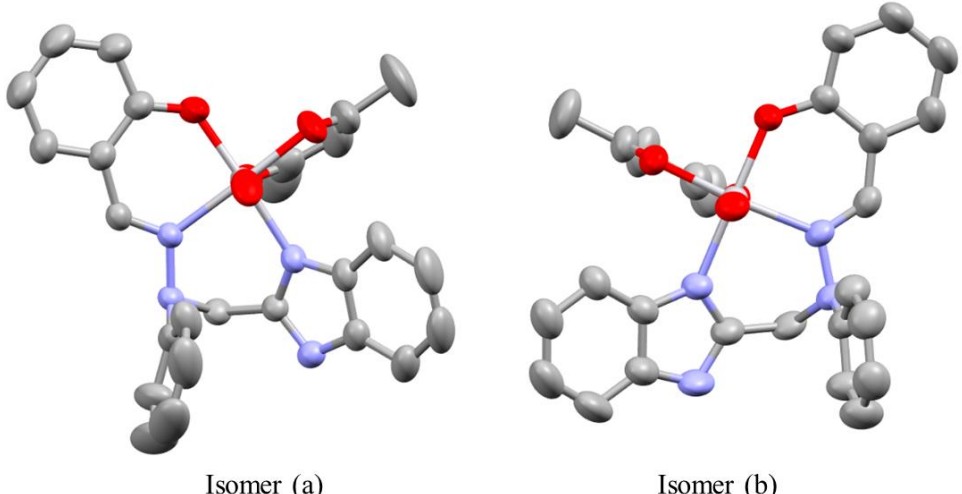

Isomer (a)                    Isomer (b)

**Figure 7.** View of enantiomers (**a**,**b**) present in the asymmetric unit of the crystal packing in complex **1**, drawn as mirror images. Non-hydrogen atoms were drawn using 50% probability ellipsoids. Hydrogen atoms were omitted for simplicity. The drawings were made with the mercury 3.8 program.

In **I**, an intramolecular hydrogen bond was observed between O(1)···N(1) atoms, with distances of 0.82 Å for D–H, 1.95 Å for H–A, 2.673(2) Å for D–A and an angle of 146.6°. In **1**, two intermolecular hydrogen bonds were observed between N(4)···O(4) atoms, with distances of 0.88 Å for D–H, 1.95 Å for H–A, 2.8111(17) Å for D–A and an angle of 165.7° and between N(4)···O(2A) atoms, with distances of 0.88 Å for D–H, 2.65 Å for H–A, 3.1638(18) Å for D–A and an angle of 118.3 °. In the crystal packing of **1**, two intermolecular interactions with hydrogen bonds were observed when symmetry transformations were used to generate equivalent atoms (#1 x + 1, y, z). These interactions are (i) between N(8)···O(2) atoms, with distances of 0.88 Å for D–H, 2.29 Å for H–A, 3.126(2) Å for D–A and an angle of 157.6° and (ii) between N(8)···O(4A) atoms, with distances of 0.88 Å for D–H, 2.42 Å for H–A, 3.065(2) Å for D–A and an angle of 130.2°.

In **1**, the coordination environment around the vanadium metal center corresponds to a distorted octahedral geometry. The trans position, with respect to the terminal oxide atom, is occupied for one oxygen atom of the acetylacetonate ligand, with a distance V(1)–O(3A) of 2.1303(14) Å, and a distance to the terminal oxygen atom V(1)–O(1) of 1.6002(15) Å (see Table S2). The distances of the metal center to donor atoms of the tridentate ligand, O(2), N(1) and N(3) are also given in Table S2, and they are similar to those seen in other vanadium complexes of the same type [73]. The equatorial plane formed for N(1)–N(2)–O(2)–O(4A) atoms in **1** (a) presents a mean deviation from planarity of 0.0143(8) Å, and the

equatorial plane formed for N(5)–N(7)–O(2)–O(2A) in **1** (b) presents a mean deviation from planarity of 0.0602(7) Å, with the vanadium atoms located out of the plane at a distance of 0,1986(8) Å and 0.2388(7) Å, respectively. The distances to apical atoms (for example, in **1** (a), the distances to V(1)–O(1) and V(1)–O(3A)) are very different from each other, which influences the distortion of the octahedron.

### 2.5. IR Spectral Study

The IR spectra of the ligands show a sharp band at 1628 cm$^{-1}$ (in **I**) and 1594 cm$^{-1}$ (in **II**) due to ν(C=N) stretch [74,75]. Complexes **1** and **2** exhibit (Figure 8) this band at a lower wavenumber and appear at 1608 (in **1**) and 1582 cm$^{-1}$ (in **2**), confirming the coordination of azomethine/ring nitrogen to vanadium. The presence of a few weak intensity bands covering the 2600–2900 cm$^{-1}$ region in ligands **I** and **II** as well as in complexes **1** and **2** are due to the CH$_2$/NH groups. In addition, complexes **1** and **2** show one sharp band at 947 and 926 cm$^{-1}$, respectively, arising due to ν(V=O) stretch, which further confirms the successful formation of complexes.

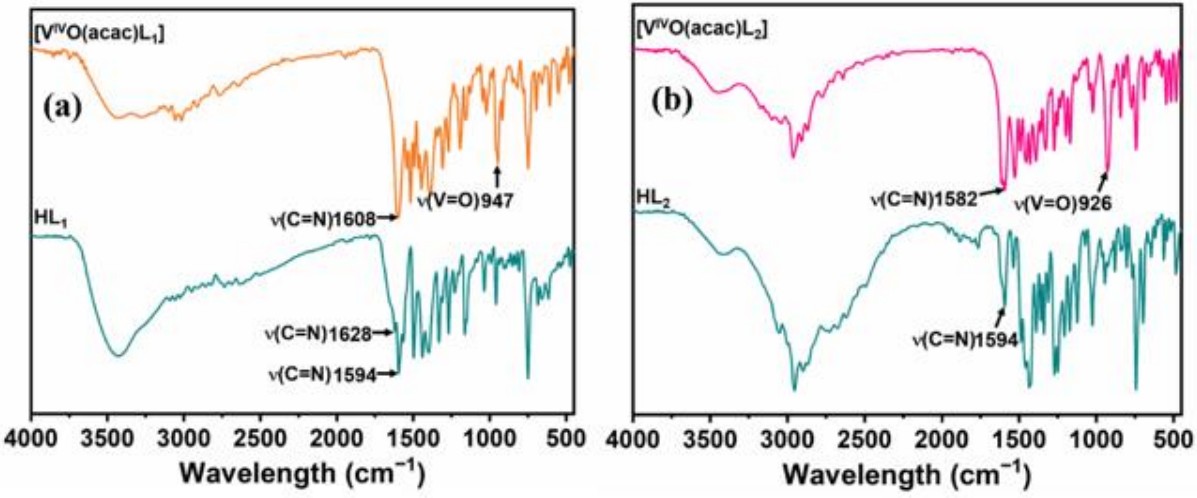

**Figure 8.** FT-IR spectra of (**a**) HL$_1$ and [V$^{IV}$O(acac)L$_1$], and (**b**) HL$_2$ and [V$^{IV}$O(acac)L$_2$].

Figure 9 compares the FT-IR spectra of chloromethylated polystyrene (PS–Cl) and polymer-supported complexes. The intensity of most IR bands of the complexes is generally weaker compared to the corresponding homogeneous complexes. The sharp band appearing at 1612 cm$^{-1}$ due to ν(C=N) stretch is broadened in complexes **3** and **4** due to its merger with ν(C=C) present in the polymer backbone. No additional band in the ca. 3000 cm$^{-1}$ region compared to polystyrene suggests the presence of only the methylene group and the absence of the –NH group. The absence of a band due to the –NH group is due to its involvement in covalent bond formation with the polymer backbone. In addition, one weak band at 981 cm$^{-1}$ in **3** and 979 cm$^{-1}$ in **4** is indicative of the presence of the V=O bond [75]. A weak band due to (C–Cl) is still present in both heterogeneous complexes at ca. 700 cm$^{-1}$ indicating that part of the PS–Cl group of the polymer is still available in these complexes.

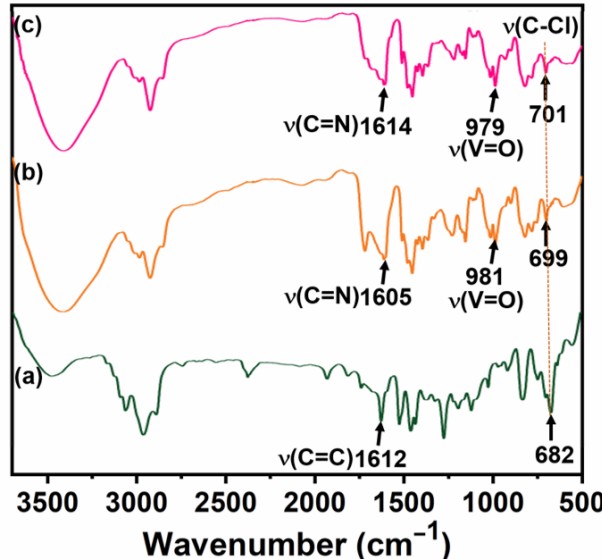

**Figure 9.** FT-IR spectra of (**a**) PS–Cl, (**b**) [$V^{IV}$(acac)$L_1$]@PS (**3**) and (**c**) [$V^{IV}$(acac)$L_2$]@PS (**4**).

### 2.6. UV-Visible Spectral Study

The electronic absorption spectra of the ligands and complexes (Figure 10) were recorded in DMSO, and the absorption maxima along with their extinction coefficients are presented in Table 1. The UV region of both ligands exhibits a set of four medium-intensity bands at 258, 277, 283 and 299 nm. The first and last bands are assigned due to the $\sigma \rightarrow \sigma^*$ and $\pi \rightarrow \pi^*$ transitions, respectively, while the other two bands are characteristics of the benzimidazole group [75]. An intense band at ca. 345 nm at the junction of the UV-visible region is assigned due to the $n \rightarrow \pi^*$ transition. All these bands can also be visualized in complexes with slight shifting and changes in intensity. In addition, both complexes exhibit a new band at ca. 405 nm (specifically 403 nm in **1** and 414 nm in **2**) possibly due to the $d \rightarrow d$ transition. The ligand-to-metal charge transfer (LMCT) from a phenolate oxygen atom to an empty d-orbital of a vanadium atom has not been observed clearly, and it seems that it is merging with the $n \rightarrow \pi^*$ transition. No other band in the visible region with a higher concentration has been observed.

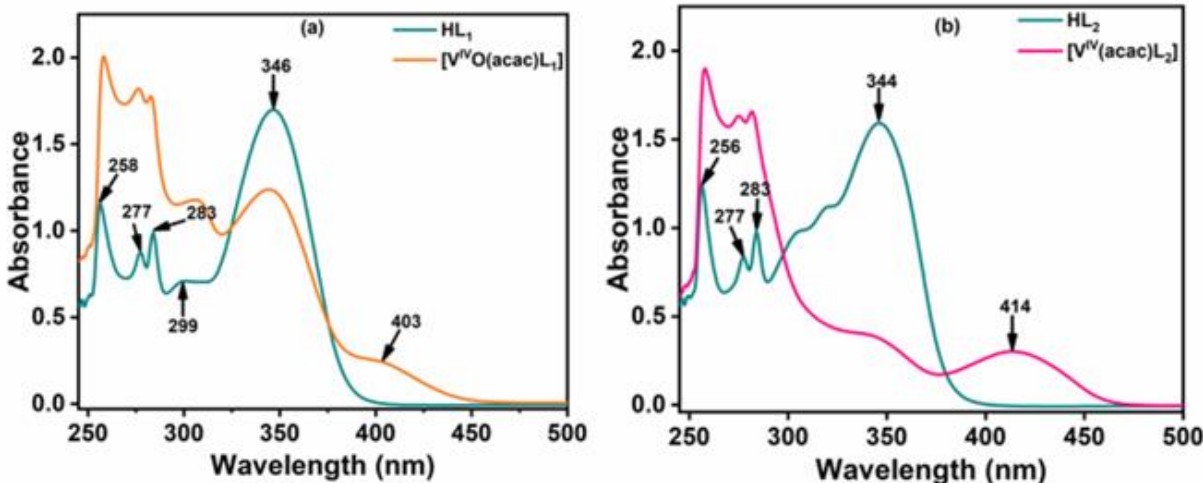

**Figure 10.** UV-visible spectra of (**a**) $HL_1$ (**I**) and [$V^{IV}$(acac)$L_1$] (**1**), and (**b**) UV-visible spectra of $HL_2$ (**II**) and [$V^{IV}$O(acac)$L_2$] (**2**).

**Table 1.** UV-visible spectra of the ligand, homogeneous and heterogeneous complexes.

| Entry | Compound | Solvent | $\lambda_{max}$/nm ($\varepsilon$/M$^{-1}$ cm$^{-1}$) |
|:---:|:---:|:---:|:---:|
| 1 | HL$_1$ (**I**) | DMSO | 258 (1.80 × 10$^3$), 277 (1.35 × 10$^3$), 283 (1.52 × 10$^3$), 299 (1.10 × 10$^3$), 346 (2.65 × 10$^3$) |
| 2 | HL$_2$ (**II**) | DMSO | 256 (1.96 × 10$^3$), 277 (1.31 × 10$^3$), 283 (1.54 × 10$^3$), 344 (2.53 × 10$^3$) |
| 3 | [V$^{IV}$O(acac)L$_1$] (**1**) | DMSO | 258 (3.06 × 10$^3$), 277 (2.79 × 10$^3$), 283 (2.71 × 10$^3$), 306 (1.81× 10$^3$), 346 (1.91 × 10$^3$), 403 (0.38 × 10$^3$) |
| 4 | [V$^{IV}$O(acac)L$_2$] (**2**) | DMSO | 258 (2.96 × 10$^3$), 275 (2.54 × 10$^3$), 282 (2.58 × 10$^3$), 342 (0.61 × 10$^3$), 414 (0.47 × 10$^3$) |
| 5 | [V$^{IV}$(acac)L$_1$]@PS (**3**) | Nujol | 230, 272, 353, 409 |
| 6 | [V$^{IV}$(acac)L$_2$]@PS (**4**) | Nujol | 232, 289, 356, 409 |

The UV-visible spectra of the polymer-supported complexes **3** and **4** along with polystyrene methyl chloride (PS-Cl) were recorded after dispersing them in Nujol (Figure 11). The UV-visible spectra of both complexes exhibit similar bands to that of analogous homogeneous complexes **1** and **2** but with a slight shift in the wavelengths and lower intensities. The absence of characteristic bands due to the benzimidazole group is in the line of the covalent bonding of nitrogen of the benzimidazole to the polymer after the removal of hydrogen. Again, the LMCT transition is possibly merging with the n → π* transition and appears at ca. 360 nm, while the band due to the d → d transition appears at ca. 409 nm. All these confirm the integrity of complexes **3** and **4** supported on the polymer matrix. Table 1 presents spectral data of all compounds.

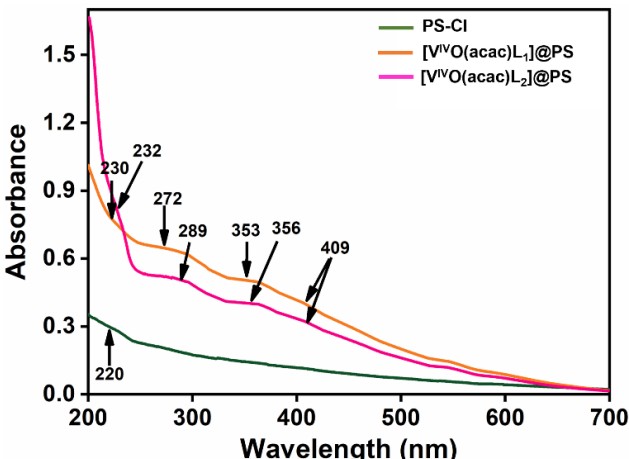

**Figure 11.** UV-visible spectra of PS-Cl, [V$^{IV}$(acac)L$_1$]@PS (**3**) and [V$^{IV}$(acac)L$_2$]@PS (**4**) recorded in Nujol.

*2.7. EPR Spectral Study*

The X-band 1$^{st}$ derivative EPR spectra of complexes, [V$^{IV}$O(acac)(L$_1$)] (**1**), [V$^{IV}$O(acac)(L$_2$)] (**2**), [V$^{IV}$O(acac)L$_1$]@PS (**3**) and [V$^{IV}$O(acac)L$_2$]@PS (**4**) were recorded in DMSO at 100 K, and the EPR spectra of these four complexes are reproduced in Figure 12. The EPR spectra of both the homogeneous and polymer-supported heterogeneous oxidovanadium(IV) complexes show eight lines without any noticeable broadening, and this ensures the presence of a V(IV) center in these complexes. These spectra confirm that the unpaired single electron of the V(IV) center is localized only on the metal center. The spin Hamiltonian parameters (Table 2) for all four complexes were calculated from the obtained spectra, and they are in the range of normal [V$^{IV}$O] octahedral complexes having N and O coordination [76]. The

high value of [$A_z$] (Table 2) is possibly due to the longer V–O (of acac ligand) bond trans to the axial V=O bond.

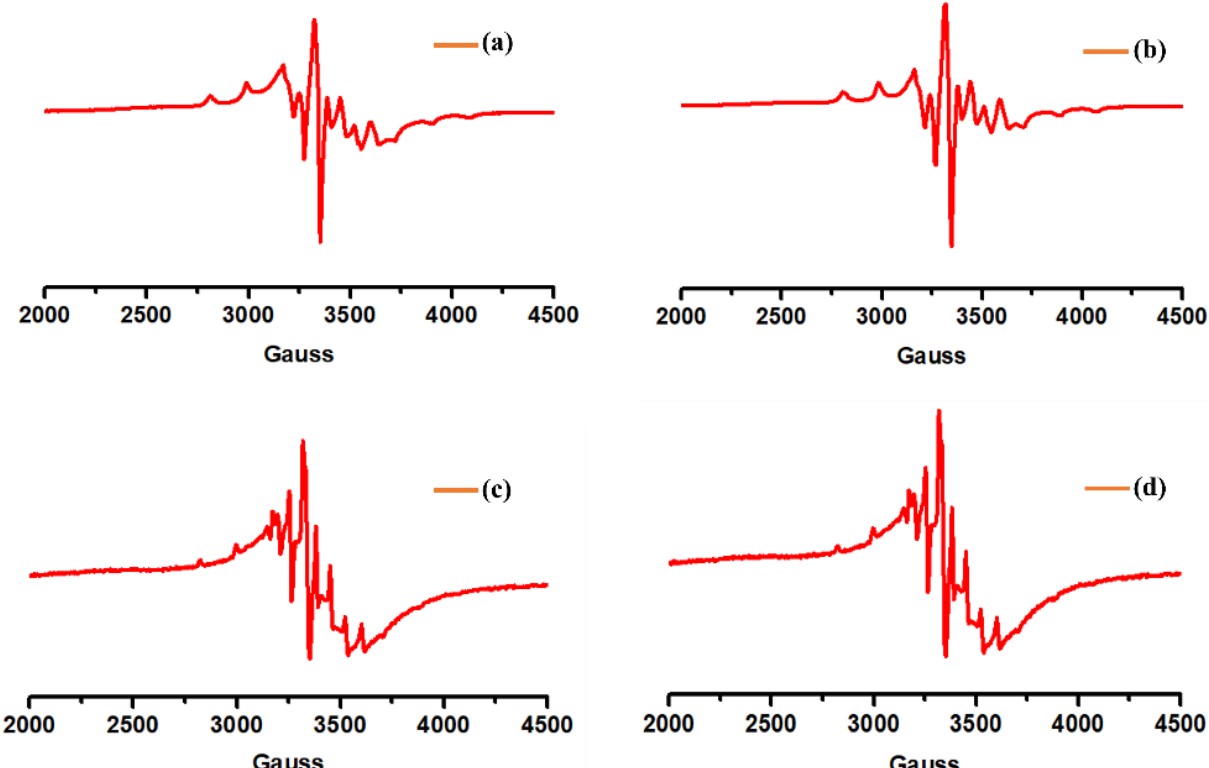

**Figure 12.** X-band EPR spectra of (**a**) complex **1**, (**b**) complex **2**, (**c**) complex **3** and (**d**) complex **4** recorded in DMSO at 100 K.

**Table 2.** Spin Hamiltonian Parameters Obtained from the EPR Spectra of Complexes **1**, **2**, **3** and **4** Recorded in DMSO at 100 K.

| Compound | $g_x, g_y$ | $\lvert A_x \rvert, \lvert A_y \rvert$ ($\times 10^{-4}$ cm$^{-1}$) | $g_z$ | $\lvert A_z \rvert$ ($\times 10^{-4}$ cm$^{-1}$) |
|---|---|---|---|---|
| [V$^{IV}$O(acac)(L$_1$)] (**1**) | 1.970 | 89.9 | 1.945 | 184.4 |
| [V$^{IV}$O(acac)(L$_2$)] (**2**) | 1.968 | 86.6 | 1.951 | 181.2 |
| [V$^{IV}$O(acac)L$_1$]@PS (**3**) | 1.971 | 72.71 | 1.945 | 170.4 |
| [V$^{IV}$O(acac)L$_2$]@PS (**4**) | 1.969 | 70.74 | 1.952 | 168.38 |

### 2.8. Powder-X-ray Diffraction Analysis

The P-XRD patterns of the prepared homogeneous complexes **1** and **2** are presented in Figure 13. The theoretical plot of complex **1** was generated using the CIF file in the Mercury software, and the generated plot was compared with the corresponding experimentally obtained P-XRD plot (Figure 13a). It is clear that the theoretical and experimental plots match well. This observation suggests that powder form of the complex retains its structural similarity. A similar pattern was also observed in the theoretical and experimental plots of complex **2** (Figure 13b). Due to the spherical shape of polystyrene beads of heterogeneous complexes **3** and **4**, no attempt was made to obtain their P-XRD patterns.

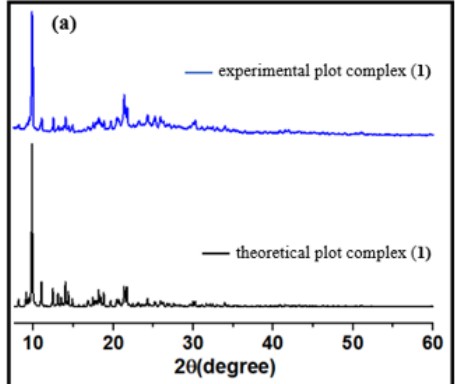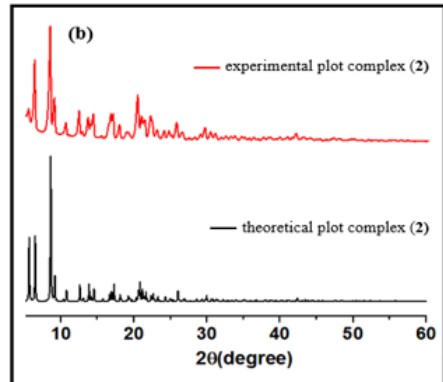

**Figure 13.** (**a**) Theoretical and experimental P-XRD patterns of complex **1**, and (**b**) theoretical and experimental P-XRD patterns of complex **2**.

### 2.9. Field Emission-Scanning Electron Microscopy (FE-SEM) and Energy Dispersive X-ray Analysis (EDS)

The FE-SEM analysis of neat polystyrene beads and polymer-supported complexes was carried out to observe the morphological changes that occurred before and after the complex formation [77]. Figure 14 shows the FE-SEM images and elemental mapping of chloromethylated polystyrene. The images of beads of polymer-supported complexes along with their elemental mapping are shown in Figure 15(a-i–b-vi). The high-resolution images show slight smoothening in the surface of polymeric beads after the immobilization of complexes. The energy dispersive X-ray analysis of the polymer-supported complexes estimates a vanadium content of 10.7 wt% in **3** and 10.2 wt% in **4**. Thus, elemental mapping further confirms the successful immobilization of both vanadium complexes onto chloromethylated polystyrene. However, since the signal due to chlorine has still been detected in the elemental mapping analysis of both supported complexes, part of the sites seems free from covalent bonding with metal complexes.

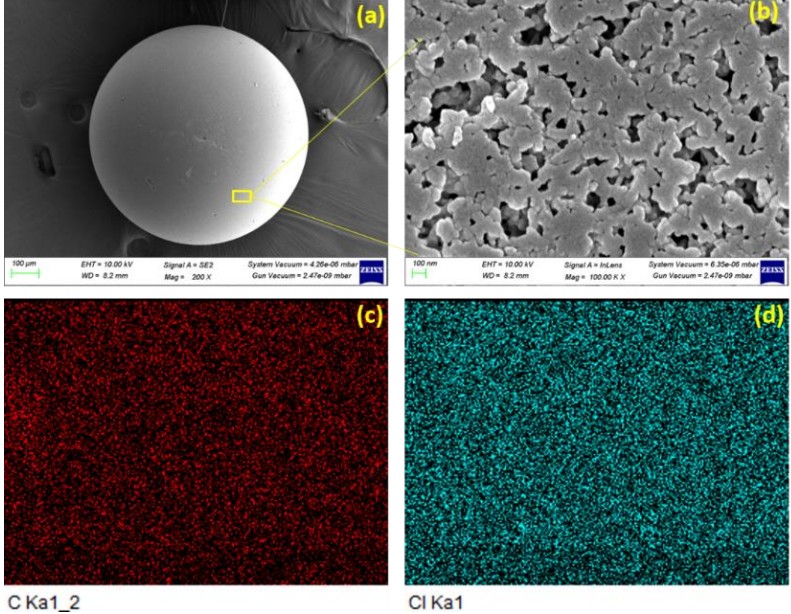

**Figure 14.** (**a**) FE-SEM image of chloromethylated polystyrene bead, (**b**) high-resolution FE-SEM image of chloromethylated polystyrene bead, (**c**) elemental mapping of carbon in chloromethylated polystyrene bead and (**d**) elemental mapping of chlorine in chloromethylated polystyrene bead.

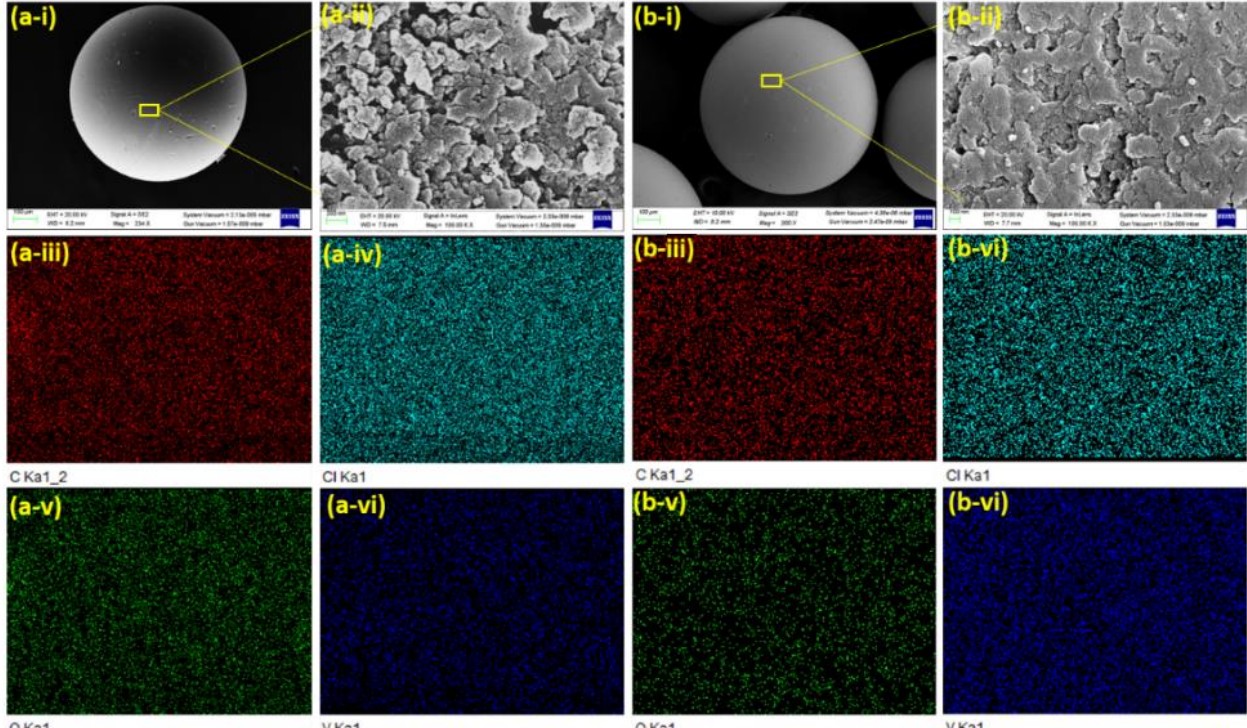

**Figure 15.** (**a-i**) FE-SEM image of **3**, (**a-ii**) high-resolution FE-SEM image of **3**, (**a-iii**) elemental mapping of carbon in **3**, (**a-iv**) elemental mapping of chlorine in **3**, (**a-v**) elemental mapping of oxygen in **3** and (**a-vi**) elemental mapping of vanadium in **3**. (**b-i**) FE-SEM image of **4**, (**b-ii**) high-resolution FE-SEM image of **4**, (**b-iii**) elemental mapping of carbon in **4**, (**b-iv**) elemental mapping of chlorine in **4**, (**b-v**) elemental mapping of oxygen in **4** and (**b-vi**) elemental mapping of vanadium in **4**.

### 2.10. Atomic Force Microscopic Study

Figure 16 shows the atomic force micrographs of pure chloromethylated polystyrene beads PS–Cl and polymer-supported heterogeneous catalysts **3** and **4**. These micrographs were analyzed, and it was observed that the average surface roughness/nm was higher for pure chloromethylated polystyrene beads PS-Cl and that it decreased in supported heterogeneous complexes **3** and **4**. The average surface roughness of the supported complexes and pure beads of PS-Cl are summarized in Table 3. The lowering of the surface roughness in supported complex **3** and **4** suggests the successful immobilization of metal complexes onto polymer beads [78–80].

**Table 3.** AFM image parameters of the polymer beads in neat and supported complexes.

| Sl. No. | Compound | Size ($\mu m^2$) | Average Surface Roughness (nm) [a] |
|---------|----------|------------------|-----------------------------------|
| 1 | PS-Cl | $10 \times 10$ | 45.1 |
| 2 | $[V^{IV}O(acac)L_1]$@PS (**3**) | $10 \times 10$ | 16.1 |
| 3 | $[V^{IV}O(acac)L_2]$@PS (**4**) | $10 \times 10$ | 13.1 |
| 4 | $[V^{IV}O(acac)L_1]$@PS (**3**) (recycled) | $10 \times 10$ | 16.2 |

[a] Average of five measurements for each bead is reported.

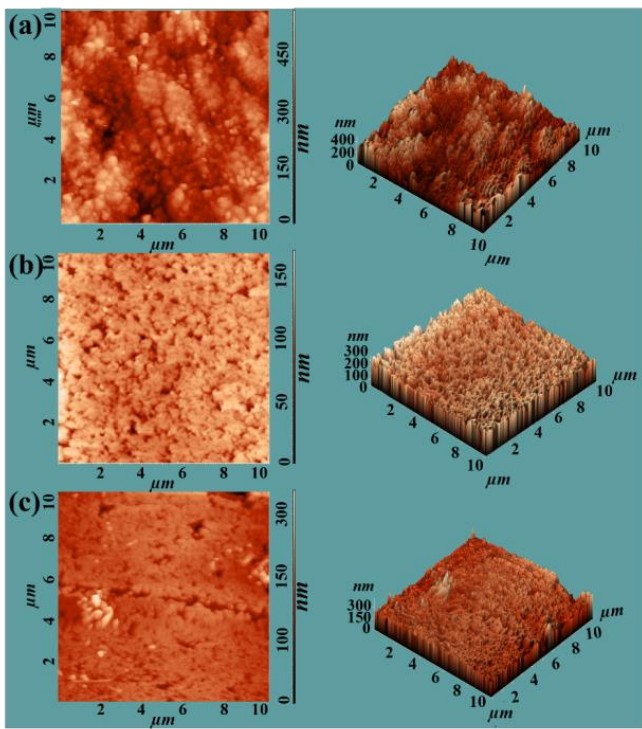

**Figure 16.** Atomic force micrograph 2D views (left) and respective 3D views (right) of (**a**) chloromethylated polystyrene PS–Cl, (**b**) [$V^{IV}O(acac)L_1$]@PS (**3**) and (**c**) [$V^{IV}O(acac)L_1$]@PS (**4**).

*2.11. Catalytic Activity Study: Synthesis of 2,4,5-Triphenyl-1H-Imidazole and Its Derivatives*

The catalytic potential of homogeneous as well as polymer-supported heterogeneous vanadium complexes has been explored for the multicomponent reaction (MCR) for the synthesis of lophine (2,4,5-triphenyl-1*H*-imidazole) and its derivatives. The MCR of benzil, ammonium acetate and different aromatic aldehydes was performed using complex **3** as a representative catalyst under different reaction conditions.

Initially, the effect of various solvents (e.g., EtOH, MeCN, THF, CHCl₃, EtOAc, water and H₂O/EtOH (1:1)) (10 mL) was screened for a fixed amount of benzil (1.050 g, 5 mmol), NH₄OAc (1.150 g, 15 mmol) and benzaldehyde (0.637 g, 6 mmol), and the reaction was carried out at 80 °C (or reflux temperature of the solvent, whichever was applicable). Amongst the solvents selected, the performance of EtOH was found to be most suitable (Figure 17a) where a 91% yield of the product was noted in 30 min at reflux temperature, while lowering the temperature (Figure 17b) resulted in a poor yield. Therefore, keeping the above factors in mind, the amount of catalyst **3** was also varied (0.005, 0.010, 0.015 and 0.020 g) under the above suitable conditions and its effect was noted. Table 4 summarizes all reaction conditions and the yield of the product under a particular set of reactions. From different reaction conditions and varied parameters, it was concluded that 0.015 g of catalyst loading in 10 mL EtOH at its reflux temperature was sufficient enough to give a maximum of 91% yield in 30 min (entry no. 12 of Table 4). This reaction was also performed with heterogeneous catalyst **4** as well as homogeneous complexes **1** and **2** considering their same mole percent based on the metal content. All catalysts have extremely good potential to give more than 92% yield of lophine. Without using catalyst **3**, only 46% of the product was achieved in 3 h of reaction time under above stated optimized reaction conditions.

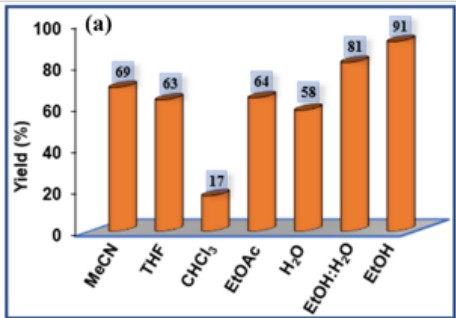 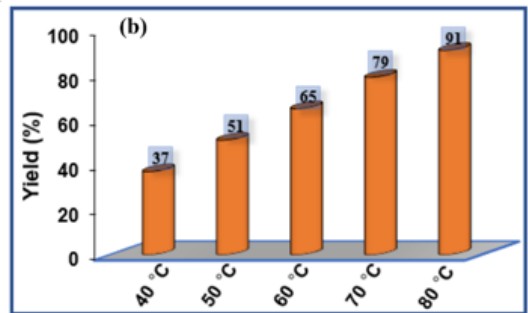

**Figure 17.** (**a**) Effect of the solvent on the yield of 2,4,5-triphenyl-1*H*-imidazole in one-pot MCR. (**b**) Effect of temperature on the yield of 2,4,5-triphenyl-1*H*-imidazole in one-pot MCR. Multi component reagents: benzil (1.050 g, 5 mmol), benzaldehyde (0.637 g, 6 mmol) and NH$_4$OAc (1.150 g, 15 mmol). Reaction conditions: catalyst 3 (0.015 g), EtOH (10 mL) at its reflux temperature and reaction time (30 min).

**Table 4.** Screening of V-catalyzed one-pot-three-component reaction for the synthesis of 2,4,5-triphenyl-1*H*-imidazole in different reaction conditions using complex **3** as a representative catalyst (component: benzil (1.050 g, 5 mmol), benzaldehyde (0.637 g, 6 mmol) and NH$_4$OAc (1.150 g, 15 mmol)).

| Entry | Catalyst (g) | Solvent (10 mL) | Time (min) | Temp. (°C) | Yield (%) |
|---|---|---|---|---|---|
| 1 | 0.015 | MeCN | 15 | 80 | 69 |
| 2 | 0.015 | THF | 15 | Reflux | 63 |
| 3 | 0.015 | CHCl$_3$ | 15 | Reflux | 17 |
| 4 | 0.015 | EtOAc | 15 | Reflux | 64 |
| 5 | 0.015 | H$_2$O | 15 | 80 | 58 |
| 6 | 0.015 | EtOH:H$_2$O (*V/V*) | 15 | 80 | 81 |
| 7 | 0.015 | EtOH | 15 | Reflux | 89 |
| 8 | 0.015 | EtOH | 15 | 70 | 79 |
| 9 | 0.015 | EtOH | 15 | 60 | 65 |
| 10 | 0.015 | EtOH | 15 | 50 | 51 |
| 11 | 0.015 | EtOH | 15 | 40 | 37 |
| 12 [a] | 0.015 | EtOH | 30 | Reflux | 91 |
| 13 | 0.015 | EtOH | 45 | Reflux | 92 |
| 14 | 0.05 | EtOH | 60 | Reflux | 71 |
| 15 | 0.010 | EtOH | 60 | Reflux | 83 |
| 16 | 0.015 | EtOH | 60 | Reflux | 91 |
| 17 | 0.020 | EtOH | 60 | Reflux | 92 |

[a] Optimized reaction condition.

### 2.12. Scope of the MCR to Other Lophine Derivatives

This one-pot MCR model has been extended to other aromatic mono aldehydes such as *p*-methylbenzaldehyde, *p*-methoxybenzaldehyde, *p*-bromobenzaldehyde, *p*-chlorobenzaldehyde and *p*-nitrobenzaldehyde as well as bis(aldehyde) such as terepthalaldehyde, and the reaction was carried out under the above stated optimized reaction conditions. Scheme 3 summarizes the yield of all products. From the yield obtained, it is clear that aromatic aldehyde with electron withdrawing substituents produce better yield (up to 96%), while electron donating substituents produce similar or less conversion compared to benzaldehyde. Interestingly, formaldehyde also gave very good conversion of respective product (1(i) of Scheme 3)).

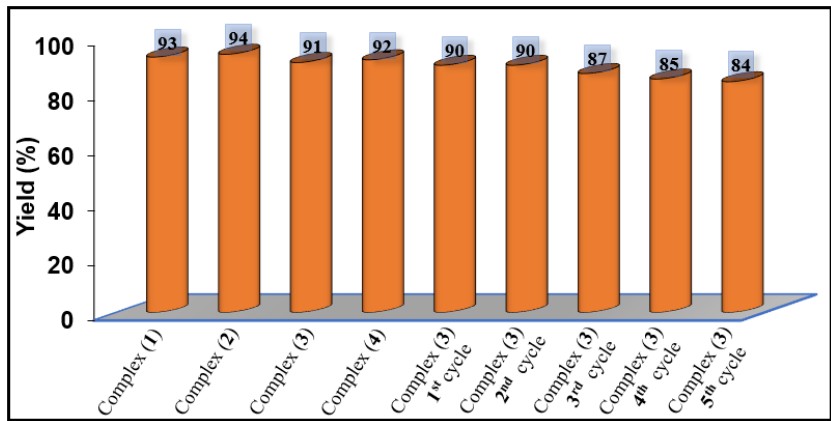

**Scheme 3.** Substrate scope for the one-pot MCRs and yield of different derivatives of 2,4,5-triphenyl-1*H*-imidazole derivatives. Multicomponent reagents: benzil (1.050 g, 5 mmol), benzaldehyde or its derivatives (6 mmol) and NH$_4$OAc (1.150 g, 15 mmol), catalyst **3** (0.015 g), EtOH (10 mL) at its reflux temperature and reaction time (30 min).

### 2.13. Regeneration of the Supported Catalyst and Study of Recyclability and Stability

The regeneration procedure for the catalyst was carried out by recovering, e.g., heterogeneous catalyst **3** from the catalytic reaction (see the experimental section for details). Since the recovery of the catalyst was not quantitative, the catalyst recovered from such two efforts was combined, and its required amount was used for the next cycle. Figure 18 presents the percent yield of catalysts **1** to **4** as fresh and of recycled **3** for five times. Only a small loss in the yield of the product even after the fifth cycle suggests its good recyclability.

**Figure 18.** Bar diagrams presenting yield by different catalysts and results of five recycles of catalyst **3** one-pot MCR for the synthesis of 2,4,5-triphenyl-1*H*-imidazole. Multicomponent reagents: benzil (1.050 g, 5 mmol), benzaldehyde (0.610 g, 6 mmol) and NH$_4$OAc (1.150 g, 15 mmol). Reaction conditions: **1** (0.62 mg), **2** (0.76 mg), **3** (15 mg) **4** (15 mg), EtOH (10 mL) at reflux temperature and reaction time (30 min).

The stability and structural integrity of the recovered catalyst **3** after the first cycle was also checked using a spectroscopic study (IR, UV-visible (Figure 19) and AFM studies (Figure 20)) as well as FE-SEM along with elemental mapping (Figure 21). Almost the

same spectra of the fresh as well as the recycled catalyst indicate the stability of the catalyst even after use. The average surface roughness of the recycled catalyst **3** only slightly increases (16.1 of fresh catalyst **3** vs. 16.2 of recycled catalyst) which further indicates a little deformation of the beads after the first cycle. While high-resolution images show almost the same smoothening of the surface of polymeric beads as was observed after the immobilization of fresh complexes, the elemental mapping of recycled catalyst **3** estimates the vanadium content of 9.85 wt%. Thus, compared to the vanadium content in the fresh catalyst (10.7 wt%), it is slightly lower, which suggests the partial loss of metal content or in the general complex from the polymer beads after the first cycle.

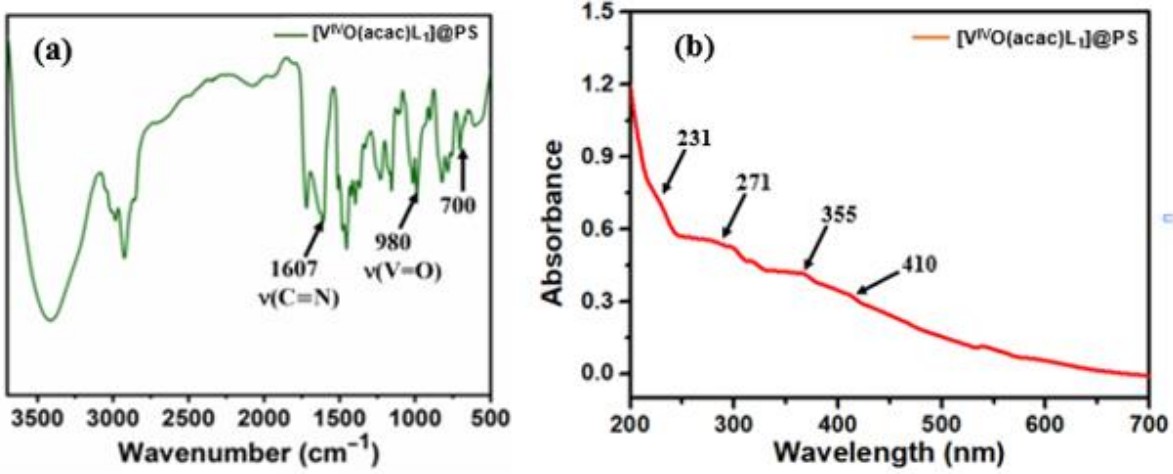

**Figure 19.** FT-IR spectrum (**a**) and UV-visible spectrum (**b**) of reused [$V^{IV}O(acac)L_1$]@PS (**3**).

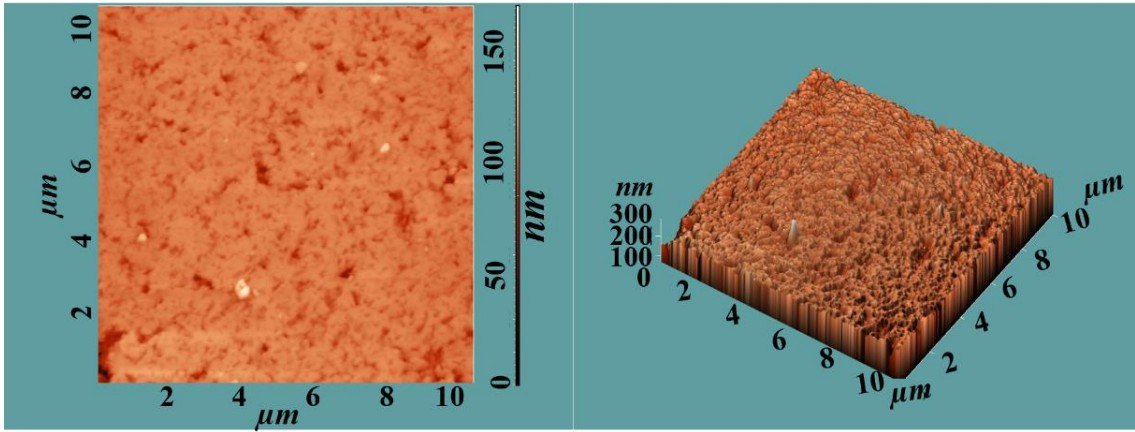

**Figure 20.** Atomic force micrograph 2D views (left) and respective 3D views (right) of recycled [$V^{IV}O(acac)L_1$]@PS (**3**).

The X-band 1st derivative EPR spectra of the regenerated catalyst [$V^{IV}O(acac)L_1$]@PS (**3**) were recorded in DMSO at 100K, and the EPR spectra of this reused complex are reproduced in Figure 22. Thus, the EPR spectrum of polymer-supported recycled [$V^{IV}O(acac)L_1$]@PS (**3**) also shows eight lines with almost the same spin Hamiltonian parameters ($g_x$, $g_y$: 1.964; $g_z$: 1.951; $|A_x|$, $|A_y|$: $69.31 \times 10^{-4}$ cm$^{-1}$; $|A_z|$: $170.89 \times 10^{-4}$ cm$^{-1}$) as observed for fresh catalyst without any noticeable broadening, and this ensures the presence of a V(IV) center in these complexes. This analysis confirms its structural integrity, stability and suitability for long-run catalytic applications. While the comparison between homogeneous and heterogeneous catalysts suggests better outcomes using homogeneous catalysts, the recyclability and stability of heterogeneous catalysts (e.g., catalyst **3**) make them better for industrial purposes.

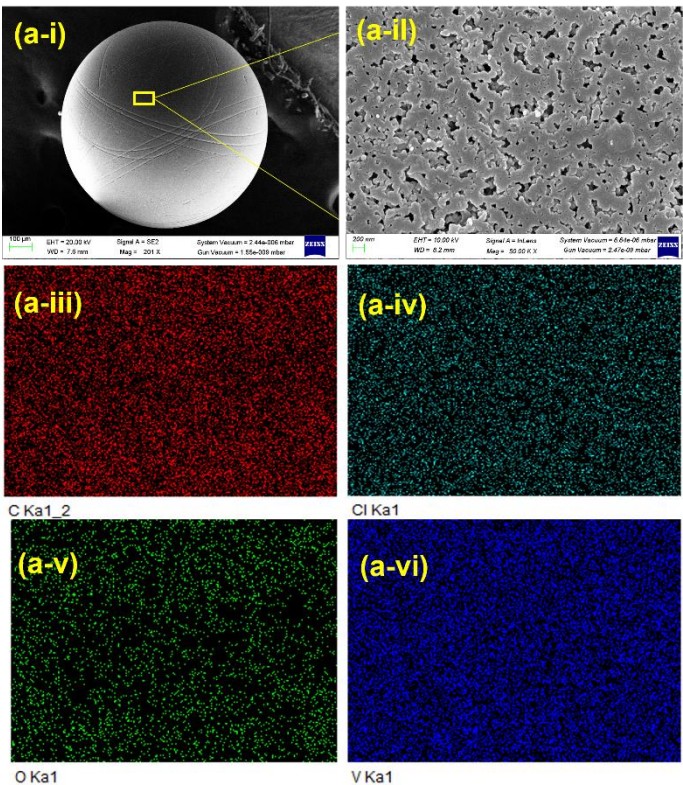

**Figure 21.** (**a-i**) FE-SEM image of recycled catalyst [$V^{IV}O(acac)L_1$]@PS (**3**), (**a-ii**) high-resolution FE-SEM image of recycled catalyst [$V^{IV}O(acac)L_1$]@PS (**3**), (**a-iii**) elemental mapping of carbon in recycled catalyst [$V^{IV}O(acac)L_1$]@PS (**3**), (**a-iv**) elemental mapping of chlorine in recycled catalyst [$V^{IV}O(acac)L_1$]@PS (**3**), (**a-v**) elemental mapping of oxygen in recycled catalyst [$V^{IV}O(acac)L_1$]@PS (**3**) and (**a-vi**) elemental mapping of vanadium in recycled catalyst [$V^{IV}O(acac)L_1$]@PS (**3**).

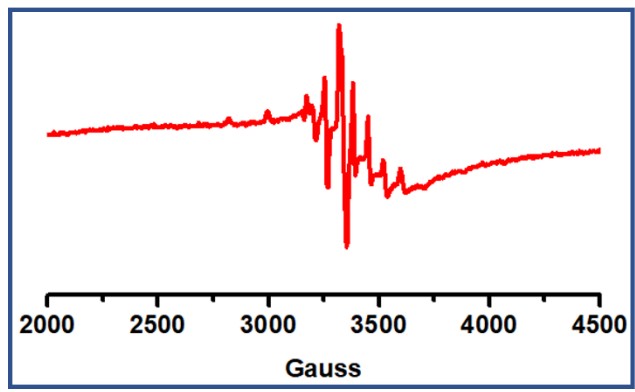

**Figure 22.** X-band EPR spectrum of reused complex [$V^{IV}O(acac)L_1$]@PS (**3**) recorded in DMSO at 100 K (for spin Hamiltonian parameters, see text).

### 2.14. Comparison of the Catalytic Efficiency of Catalyst **3** with the Literature Data

The catalytic performance of catalyst **3** compares well with the catalysts reported in the literature. Table 5 compares the reaction conditions, time and yield of 2,4,5-triphenyl-1*H*-imidazole derivatives using various catalysts reported in the literature. It was found that most of the catalysts have competing yields. While Lewis acid- or base-based catalysts required a long reaction time, most of the supported and heterogeneous catalysts required less time but a higher temperature compared to this work. The homogeneous complex as well as the polymer-supported heterogeneous catalyst reported in this work have shown excellent catalytic efficiency with a shorter reaction time and lower temperature.

**Table 5.** Comparison of the Results for the Synthesis of 2,4,5-Triphenyl-1*H*-Imidazole Derivatives using Different Catalysts.

| Entry No. | Catalyst and Conditions | Reaction Time (Minutes) | Yield (%) | Ref. |
|---|---|---|---|---|
| 1 | [Hbim] BF$_4$ (4 mmol)/solvent free/ 100 °C | 60 | 95 | [56] |
| 2 | Silica Sulfuric acid (500 mg)/ water/reflux | 240 | 73 | [57] |
| 3 | InCl$_3$·3H$_2$O/MeOH/rt | 500 | 82 | [58] |
| 4 | L-proline/MeOH/rt | 540 | 90 | [59] |
| 5 | DABCO (0.7 mol%)/*t*-BuOH/65 °C | 45 | 97 | [60] |
| 6 | Nontmorilonite/EtOH/reflux | 90 | 70 | [61] |
| 7 | Fe$_3$O$_4$-PEG-Cu/solvent free/110 °C | 30 | 98 | [62] |
| 8 | Scolecite (2 wt%)/lactic acid/160 °C | 180 | 90 | [67] |
| 9 | NFS-PMA (20 mg)/solvent free/120 °C | 20 | 94 | [63] |
| 10 | -CD-PSA (2 mol%)/solvent free/ 100 °C | 20 | 96 | [64] |
| 11 | CSNP/MWCNT@Fe$_3$O$_4$/EtOH/reflux | 60 | 86 | [65] |
| 12 | MIP Nanoreactors/solvent free/120 °C | 20 | 97 | [66] |
| 13 | LADES@MNP/solvent free/sonication | 120 | 83 | [68] |
| 14 | TBHDPB (5 mol%)/EtOH/reflux | 60 | 85 | [70] |
| 15 | CoFe$_2$O$_4$@SiO$_2$@(CH$_2$)$_3$OWO$_3$H NPs(10 mg)/solvent free/110 °C | 20 | 87 | [71] |
| 16 | V catalyst 1 (0.62 mg)/EtOH/reflux | 30 | 93 | This work |
| 17 | V catalyst 3 (15 mg)/EtOH/reflux | 30 | 91 | This work |

*2.15. Reactivity of Complex **1** with Multicomponent Reagents and a Possible Reaction Mechanism*

Encouraged by the successful catalytic results, it was important to sketch out the possible reaction mechanism for the one-pot three-component reaction for efficient synthesis of lophine derivatives. Therefore, we performed several experiments considering (**1**) as a representative. Their details and resulting possible interpretations are given here.

A solution of complex **1** (0.002 g, $3.46 \times 10^{-3}$ M) in DMSO (10 mL) was diluted three times (final concentration ($8.66 \times 10^{-5}$ M)) and then treated with one-drop portions of benzaldehyde (0.010 g, $9.4 \times 10^{-3}$ M) dissolved in 10 mL of DMSO, and the resulting spectral change was monitored using UV-visible spectroscopy. The observed changes are depicted in Figure 23. Here, the intensity of the LMCT band (405 nm) slightly decreases while that of the n → π* transition slightly increases with no change in their positions. Simultaneously, three bands appearing in the UV region change their intensities significantly and merge into one broadband and appear at ca. 290 nm (Figure 23a). These changes clearly show the interaction of benzaldehyde with the vanadium center. The above solution was then treated dropwise with an ammonium acetate solution (0.010 g, $1.29 \times 10^{-2}$ M) dissolved in 10 mL of DMSO, which shows a decrease in the intensity of the band at 345 nm, while the other two bands (290 and 405 nm) show only minor changes (Figure 23b). This is possibly due to the reaction of ammonium acetate with vanadium-bonded benzaldehyde. Treatment with a solution of complex **1** ($8.66 \times 10^{-5}$ M) separately with benzil (0.020 g, $9.52 \times 10^{-3}$ M) also resulted in a similar change in intensities for all bands (Figure 23c) as observed in the first case (i.e., Figure 23a). The only difference is the gradual change in the intensity with the addition of benzil. These changes also indicate the interaction of the vanadium center with benzil.

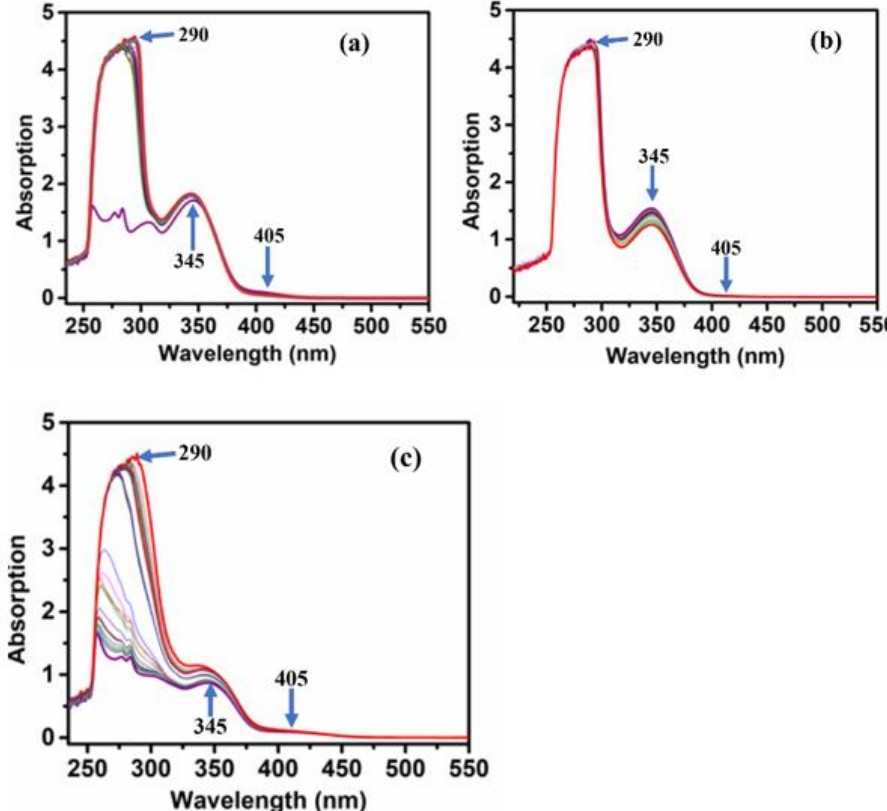

**Figure 23.** (**a**) Spectral changes observed after successive addition of a one-drop portion of benzaldehyde ($9.4 \times 10^{-3}$ M) dissolved in 10 mL of DMSO into a 10 mL DMSO solution of complex (**1**) ($8.66 \times 10^{-5}$ M) for every 2 min interval. (**b**) Spectral changes observed after the successive addition of a one-drop portion of ammonium acetate dissolved in 10 mL of DMSO ($1.29 \times 10^{-2}$ M) to a solution of (**a,c**) Spectral changes observed after the successive addition of a one-drop portion of benzil ($9.52 \times 10^{-3}$ M) to the solution of complex **1** for every 2 min interval.

### 2.16. Mechanistic Study of the Polymer-Supported Heterogeneous Oxidovanadium(IV) Catalyst in the Synthesis of Lophine Derivatives

The carbonyl group of aromatic aldehyde was proposed to be activated by coordination with the vanadium metal center of the catalysts. Ammonia ($NH_3$) generated from ammonium acetate reacted with the activated aldehyde giving rise to an imine intermediate (**I**) followed by a diamine intermediate (**II**). A similar coordination and activation of benzil resulted in an increase in electrophilicity of the C=O group, and the attack of diamine intermediate (**II**) led to the formation of intermediate (**III**), followed by intermediate (**IV**). Finally, 2,4,5-trisubstituted imidazole products were formed through the release of a water molecule and followed by the cyclization process. The coordination of aromatic aldehyde as well as benzil was supported by the UV-visible data described above (Figure 23). After completion of the reaction, the heterogeneous catalyst was separated from the reaction mixture using filtration, and the recycled catalyst was washed and reused for several runs. Hence, a possible reaction mechanism is sketched in Scheme 4. Similar reaction pathways were also proposed by Hajizadeh et al. [69] and Ahmadi et al. [72] for the synthesis of imidazole-containing heterocycles.

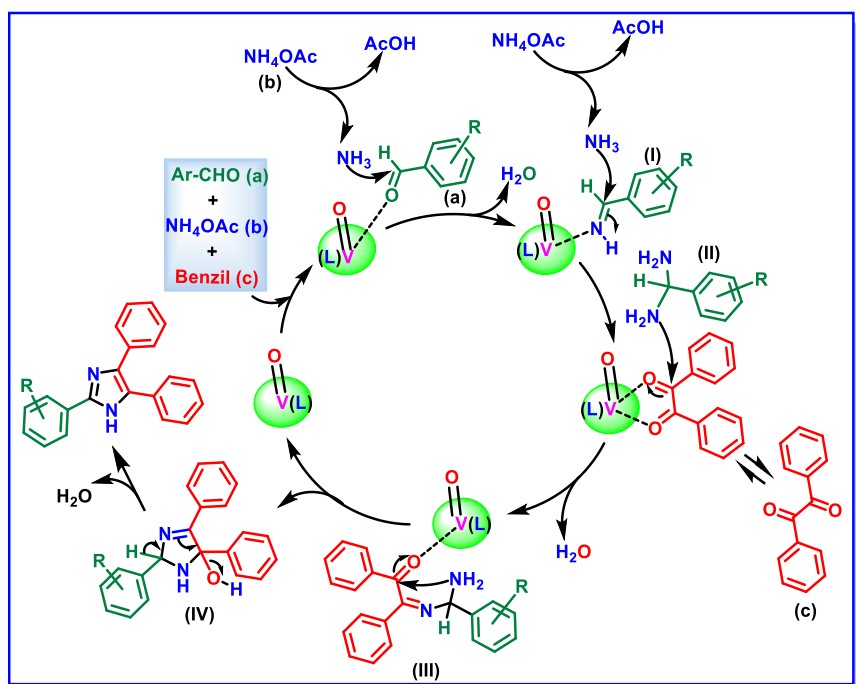

**Scheme 4.** A possible reaction mechanism of polymer-supported heterogeneous V-catalyzed one-pot MCRs for efficient synthesis of 2,4,5-triphenyl-1*H*-imidazole derivatives [59].

## 3. Experimental Section

### 3.1. Materials, Instrumentation and Characterization Procedures

Analytical reagent-grade phenylhydrazine hydrochloride, triethyl amine, benzene-1,2-diamine (Sigma-Aldrich, Milwaukee, WI, USA), acetyl acetone, benzaldehyde derivatives, salicylaldehyde, benzil, ethyl acetate, ammonium acetate and VOSO$_4$ hydrate (Sisco Research laboratories, Andheri, Mumbai, India) were used without further purification. Following the literature method [81,82], 2-Chloromethyl benzimidazole was prepared by the reaction of benzene-1,2-diamine and ethyl chloroacetate in a dilute 4M HCl solution. The vanadium precursor [V$^{IV}$O(acac)$_2$] was prepared following the literature method [83].

EPR spectra of the homogeneous as well as the heterogeneous complexes were recorded in DMSO at 100 K using a Bruker Biospin EMXmicro A200-9.5/12/S/W spectrometer, Bremen, Germany. Other instrumentation details are presented in our previously published article [84].

### 3.2. Synthesis of Ligand HL$_1$ (**I**) and HL$_2$ (**II**)

These ligands were prepared in two steps using a general procedure, and the detailed procedure for HL$_1$ (**I**) is presented here. Phenylhydrazine hydrochloride (1.446 g, 10 mmol) dissolved in 50 mL MeOH was first neutralized with triethyl amine (1.012 g, 1.4 mL, 10 mmol) while stirring the reaction mixture at room temperature for 15 min. Thereafter, salicylaldehyde (1.466 g, 12 mmol) dissolved in 10 mL of MeOH was added to the above solution, and the reaction mixture was refluxed on an oil bath for 6 h. The obtained solution was poured over crushed ice, and the precipitated white solid was filtered, washed with water, and dried over silica gel in a desiccator. In the next step, the white solid (1.06 g, 5 mmol) was dissolved in MeOH (50 mL) and treated with triethyl amine (0.506 g, 0.7 mL, 5 mmol) followed by stirring at room temperature for 15 min. To this, 2-chloromethyl benzimidazole (1.00 g, 6 mmol) was added, and the reaction mixture was stirred at room temperature for 3 h. A greyish-white solid of HL$_1$ was formed within 30 min which was filtered, washed with methanol (3 × 10 mL) and dried over silica gel in a desiccator.

Data for HL$_1$ (**I**): Yield 1.14 g (67.0%).
Data for H$_2$L$_2$ (**II**): Yield 1.39 g (61.0%).

Elemental and spectral data of both ligands are presented in the supporting information.

### 3.3. Synthesis of [V$^{IV}$O(acac)L$_1$] (**1**)

To a solution of ligand HL$_1$ (**I**) (0.343 g, 1 mmol) dissolved in 30 mL of dry MeOH was added a solution of [V$^{IV}$O(acac)$_2$] (0.320 g, 1.2 mmol) in dry methanol (20 mL), and the reaction mixture was refluxed on a water bath for 6 h. The color of the solution slowly changed to green within the first 10 min of reflux, and after ca. 20 min, it changed to brown. After 6 h, the reaction mixture was left at room temperature where a brown color sharp needle-shaped crystal along with a brown precipitate formed in the reaction flask after ca. 2 days. These were collected using filtration, washed with cold MeOH and dried in a vacuum over silica gel. Brown crystals picked up from this were found suitable for the single crystal X-ray study. Yield 0.289 g (57.0%). C$_{26}$H$_{24}$N$_4$O$_4$V (507.44) calcd: C, 61.48; H, 4.73; N, 11.03. Found: C, 61.13; H, 4.71: N, 11.01%. UV-vis (DMSO) [$\lambda_{max}$, nm ($\varepsilon$, liter mol$^{-1}$ cm$^{-1}$)]: 258 (3.06 $\times$ 10$^3$), 277 (2.79 $\times$ 10$^3$), 283 (2.71 $\times$ 10$^3$), 306 (1.81$\times$ 10$^3$), 346 (1.91 $\times$ 10$^3$), 403 (0.38 $\times$ 10$^3$). IR (KBr, $\overline{v}$/cm$^{-1}$): 1608 (C=N), 947(V=O).

### 3.4. Synthesis of [V$^{IV}$O(acac)L$_2$] (**2**)

Complex **2** was prepared using the method outlined for **1** using [V$^V$O(acac)$_2$] (0.320 g, 1.2 mmol) and HL$_2$ (0.455 g, 1 mmol) in dry MeOH. Yield 0.341 g (55.0%). C$_{34}$H$_{40}$N$_4$O$_4$V (619.66): calcd C, 65.84; H, 6.45; N, 9.03. Found: C, 65.18; H, 6.41; N, 9.01%. UV-vis (DMSO) [$\lambda_{max}$, nm ($\varepsilon$, liter mol$^{-1}$ cm$^{-1}$)]: 258 (3.06 $\times$ 10$^3$), 277 (2.79 $\times$ 10$^3$), 283 (2.71 $\times$ 10$^3$), 306 (1.81$\times$ 10$^3$), 346 (1.91 $\times$ 10$^3$), 403 (0.38 $\times$ 10$^3$). IR (KBr, $\overline{v}$/cm$^{-1}$): 1582 (C=N), 926 (V=O).

### 3.5. Synthesis of [V$^{IV}$O(acac)L$_1$]@PS (**3**)

In a 100 mL round bottom flask, 1.5 g of chloromethylated polystyrene was allowed to swell in DMF (12 mL) for 2 h. A solution of **1** (1.00 g) dissolved in DMF (10 mL) was added to the above suspension along with triethyl amine (2.66 g) and ethyl acetate (10 mL) and the final reaction mixture was heated with continuous stirring at 90 °C for 48 h. After cooling the reaction mixture at room temperature, a light-grey solid was filtered, washed with hot DMF (3 $\times$ 5 mL) followed by MeOH (3 $\times$ 5 mL) and dried in an air oven at 110 °C for overnight. The vanadium content found using MP-AES was 0.81 mmol/g. UV-vis (Nujol, $\lambda_{max}$/nm): 230, 272, 353, 409. IR (KBr, $\overline{v}$/cm$^{-1}$): 1605 (C=N), 981(V=O), 699 (C–Cl).

### 3.6. Synthesis of [V$^{IV}$O(acac)L$_2$]@PS (**4**)

Polymer-supported complex **4** was prepared similarly to the method mentioned for **3**. The vanadium content found using MP-AES was 0.82 mmol/g. UV-vis (Nujol, $\lambda_{max}$/nm): 232, 275, 322. IR (KBr, $\overline{v}$/cm$^{-1}$): 1614 (C=N), 979 (V=O), 701 (C–Cl).

### 3.7. Catalytic Activity—Synthesis of Lophine ((2,4,5-Triphenyl-1H-imidazole)) Derivatives Using the One-Pot Multicomponent Reaction

A general procedure for the synthesis of lophine (2,4,5-triphenyl-1*H*-imidazole) derivatives was applied, and the reaction was carried out in a 50 mL reaction flask. In a reaction flask, benzil (1.050 g, 5 mmol), ammonium acetate (1.150 g, 15 mmol) and benzaldehyde (0.637 g, 6 mmol) were dissolved in EtOH (10 mL). After the addition of catalyst **3** (0.015 g), the reaction mixture was refluxed on an oil bath with continuous stirring where a white precipitate started to form in the reaction mixture within 15 min. After half an hour, the reaction mixture was cooled to room temperature and the white solid was filtered, washed with CHCl$_3$ and dried. The reaction condition was optimized by varying the solvent, catalyst amount and temperature of the reaction mixture to obtain the best yield of the product. The characterization details of all reaction products are presented in the Supporting Information (yields of products, and $^1$H and $^{13}$C spectra).

## 4. Conclusions

Two heterogeneous oxidovanadium(IV) complexes, [$V^{IV}O(acac)L_1$]@PS (**3**) and [$V^{IV}O(acac)L_2$]@PS (**4**), supported on chloromethylated polystyrene and their analogous homogeneous complexes [$V^{IV}O(acac)L_1$] (**I**) and [$V^{IV}O(acac)L_2$] (**II**) of monobasic tridentate O⌢N⌢N donor ligands, $HL_1$ (**I**) and $HL_2$ (**II**), have been prepared and characterized. While spectroscopic studies characterized these complexes, a single crystal X-ray diffraction study of **I** and **1** further confirmed their structures. Eight lines of EPR spectra without any noticeable broadening support the presence of magnetically dilute and localized V(IV) centers in these complexes. Homogeneous as well as heterogeneous complexes catalyze the one-pot multicomponent (benzil, ammonium acetate and benzaldehyde) reactions for the synthesis of a biologically active imidazole core-based heterocyclic compound (2,4,5-trisubstituted-1*H*-imidazole). Extending this reaction to different derivatives of benzaldehyde and even formaldehyde produced an excellent yield of imidazole-containing products in a short reaction time under optimized reaction conditions. Catalyst **3** was recycled and reused for up to five cycles with a minimum loss of its catalytic activity. Homogeneous catalysts performed slightly better in terms of yield, but the reusability of heterogeneous catalysts makes them more useful for industrial purposes. A suitable reaction mechanism via the step-wise interaction of aldehyde and benzil has been proposed for the oxidovanadium(IV)-catalyzed synthesis of 2,4,5-trisubstituted-1*H*-imidazole.

**Supplementary Materials:** The following supporting information can be downloaded at: https://www.mdpi.com/article/10.3390/catal13030615/s1, Analytical and spectral data of ligands; Spectral data of isolated compounds (MCR products); Table S1. Selected bond lengths [Å] and angles [°] for the $HL_1$ (**I**); Table S2. Selected bond lengths [Å] and angles [°] for complex **1**; Figure S1. The $^1$H-NMR spectrum of ligand $HL_1$ (**I**) recorded in DMSO-$d_6$; Figure S2. The $^1$H-NMR spectrum of ligand $HL_2$ (**II**) recorded in DMSO-$d_6$; Figure S3. The $^{13}$C-NMR spectrum of ligand $HL_1$ (**I**) recorded in DMSO-$d_6$; Figure S4. The $^1$H-NMR spectrum of ligand $HL_2$ (**II**) recorded in DMSO-$d_6$; Figure S5. The $^1$H-NMR spectrum of **1(a)** recorded in DMSO-$d_6$; Figure S6. The $^1$H-NMR spectrum of **1(b)** recorded in DMSO-$d_6$; Figure S7. The $^1$H-NMR spectrum of **1(c)** recorded in DMSO-$d_6$; Figure S8. The $^1$H-NMR spectrum of **1(d)** recorded in DMSO-$d_6$; Figure S9. The $^1$H-NMR spectrum of **1(e)** recorded in DMSO-$d_6$; Figure S10. The $^1$H-NMR spectrum of **1(f)** recorded in DMSO-$d_6$; Figure S11. The $^1$H-NMR spectrum of **1(g)** recorded in DMSO-$d_6$; Figure S12. The $^1$H-NMR spectrum of **1(h)** recorded in DMSO-$d_6$; Figure S13. The $^1$H-NMR spectrum of **1(i)** recorded in DMSO-$d_6$; Figure S14. The $^{13}$C-NMR spectrum of **1(a)** recorded in DMSO-$d_6$; Figure S15. The $^{13}$C-NMR spectrum of **1(b)** recorded in DMSO-$d_6$; Figure S16. The $^{13}$C-NMR spectrum of **1(c)** recorded in DMSO-$d_6$; Figure S17. The $^{13}$C-NMR spectrum of **1(d)** recorded in DMSO-$d_6$; Figure S18. The $^{13}$C-NMR spectrum of **1(e)** recorded in DMSO-$d_6$; Figure S19. The $^{13}$C-NMR spectrum of **1(f)** recorded in DMSO-$d_6$; Figure S20. The $^{13}$C-NMR spectrum of **1(g)** recorded in DMSO-$d_6$; Figure S21. The $^{13}$C-NMR spectrum of **1(h)** recorded in DMSO-$d_6$; Figure S22. The $^{13}$C-NMR spectrum of **1(i)** recorded in DMSO-$d_6$.

**Author Contributions:** Conceptualization, M.R.M.; methodology, M.N. and A.P.; software, M.R.M., M.N. and A.P.; formal analysis, M.N. and A.P.; investigation, M.N. and A.P.; writing—original draft preparation, M.N. and A.P.; writing—review and editing, M.R.M. and K.G.; visualization, M.N. and A.P.; single crystal X-ray part: F.A.; supervision, M.R.M. and K.G. project administration, M.R.M.; funding acquisition, M.R.M. All authors have read and agreed to the published version of the manuscript.

**Funding:** This research was funded by the Science and Engineering Research Council (CRG/2018/000182), the Department of Science and Technology, New Delhi, the Government of India.

**Data Availability Statement:** The datasets generated during and/or analyzed during the current study are available from the corresponding author on request.

**Acknowledgments:** M.R.M. thanks the Science and Engineering Research Council, the Department of Science and Technology, the Government of India, New Delhi, for financial support for this work. M.N. and A.P. are thankful to the Council of Scientific and Industrial Research, New Delhi, for a Junior Research Fellowship. The 500 MHz NMR used for study was purchased from the DST-FIST grant by the department.

**Conflicts of Interest:** There are no conflict to declare.

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
