# Peer review of "Polymer-Supported Oxidovanadium(IV) Complexes and Their Catalytic Applications in One-Pot Multicomponent Reactions Producing Biologically Active 2,4,5-Trisubstituted-1H-imidazoles"

_catalysts, doi:10.3390/catal13030615_

Round 1

Reviewer 1 Report

1. The document requires thorough professional proofreading in terms of language improve the quality of the paper (to ensure errors such as basic grammar, style  and to correct generalized discourse). In many sentences, the wrong words were used, and as a result, self-contradictory sentences were created (the whole publication needs to be corrected)

2. The NMR spectra should be added to the supplementary materials

3. Fig. 116. the process conditions for graphs a and b should be separated

4.Figure 20. FE-SEM should be presented in comparison with results obtained for fresh catalyst.

5.All spectral and elementary data presented in the experimental section should be moved to the supplementary materials

Author Response

Comments and Suggestions for Authors

1. The document requires thorough professional proofreading in terms of language improve the quality of the paper (to ensure errors such as basic grammar, style and to correct generalized discourse). In many sentences, the wrong words were used, and as a result, self-contradictory sentences were created (the whole publication needs to be corrected)

Reply: We have tried to remove all mistakes and typos including grammar.

2. The NMR spectra should be added to the supplementary materials

Reply: NMR spectra of ligands and catalytic products have been added to supplementary materials.

3. Fig. 16. the process conditions for graphs a and b should be separated.

Reply: As suggested, the process conditions for graphs a and b have been separated now.

4.Figure 20. FE-SEM should be presented in comparison with results obtained for fresh catalyst.

Reply: Additional text has been included while comparing the FE-SEM of recycled catalyst with the fresh one.

5.All spectral and elementary data presented in the experimental section should be moved to the supplementary materials

Reply: All spectral and elementary data pertaining to ligands and catalytic reaction products have been shifted to supporting materials.

Reviewer 2 Report

Authors in the manuscript no. catalysts-2248895 entitled "Polymer-supported oxidovanadium(IV) complexes and their catalytic applications in one-pot-multicomponent reactions producing biologically active 2,4,5-trisubstituted-1H-imidazoles" have described polymer-supported oxidovanadium(IV) complexes and their catalytic applications in one-pot-multicomponent reactions producing biologically active 2,4,5-trisubstituted-1H-imidazoles. I read this work with interest. They have improved some of the reaction/applications conditions in this work. Just there is still a few lacking in the present manuscript especially in comparison of their results on green MCR protocols of the imidazoles reactions. Anyway, it is strongly recommended for publication in this journal after the following minor revisions to improve its quality before publication:

I.        In imidazole protocols study like this work, it is recommended them to explain in text the importance and relationship of the obtained activity as same as their synergistic effects on their specific structural roles.

II.        There are diverse reports in the literature on the methodologies of the aim raw materials, which mainly have used as the same strategy. They should be mentioned as tabulated and the differences between the results already published may be discussed.

III.        The manuscript is relatively well written and the conditions optimization observation noteworthy. Overall, the recent MCR protocols of the imidazoles reactions and green nanocatalysts pertinent and best texts as useful reports in the field are the following: Mater. Sci. Engin. C 2020, 109, 110502; Microporous Mesoporous Mater. 2018, 259, 46; etc. to improve their literature survey.

IV.        The quality and presentation of the figures and schemes need to be improved and well-re-drawn.

V.        MS is recommended to be re-checked in a standard format of this journal.

VI.        There are a few grammatical and typo errors in the text that need to be re-checked and corrected more carefully.

In conclusion, I hope that they find my comments constructive and I am eagerly looking forward to re-review their carefully revised and improved version.

Author Response

Comments and Suggestions for Authors

Authors in the manuscript no. catalysts-2248895 entitled "Polymer-supported oxidovanadium(IV) complexes and their catalytic applications in one-pot-multicomponent reactions producing biologically active 2,4,5-trisubstituted-1H-imidazoles" have described polymer-supported oxidovanadium(IV) complexes and their catalytic applications in one-pot-multicomponent reactions producing biologically active 2,4,5-trisubstituted-1H-imidazoles. I read this work with interest. They have improved some of the reaction/applications conditions in this work. Just there is still a few lacking in the present manuscript especially in comparison of their results on green MCR protocols of the imidazoles reactions. Anyway, it is strongly recommended for publication in this journal after the following minor revisions to improve its quality before publication:

I.  In imidazole protocols study like this work, it is recommended them to explain in text the importance and relationship of the obtained activity as same as their synergistic effects on their specific structural roles.

Reply: Few lines have been modified in the introduction considering this comment. Further, conclusion has also considered this comment.

II.        There are diverse reports in the literature on the methodologies of the aim raw materials, which mainly have used as the same strategy. They should be mentioned as tabulated and the differences between the results already published may be discussed.

Reply: As suggested, comparison from literature data along with text has been included in the revised manuscript.

III.        The manuscript is relatively well written and the conditions optimization observation noteworthy. Overall, the recent MCR protocols of the imidazoles reactions and green nanocatalysts pertinent and best texts as useful reports in the field are the following: Mater. Sci. Engin. C 2020, 109, 110502; Microporous Mesoporous Mater. 2018, 259, 46; etc. to improve their literature survey.

Reply: We appreciate the reviewer’s remark and suggestions to include above two references which are related to multicomponent Hantzsch reaction. However, these two references are related to dihydropyrimidinones/hydroquinoline and are not directly related to MCR imidazole synthesis. Therefore, we feel these are not much relevant to include in the present paper. However, reviewer may see that we have improved the introduction part.

IV.        The quality and presentation of the figures and schemes need to be improved and well-re-drawn.

Reply: Some of the figures which also felt having poor quality, have been re-drawn and included in the revised manuscript.

V.        MS is recommended to be re-checked in a standard format of this journal.

Reply: Now standard format has been used and whole manuscript has been checked for their possible grammar.

VI.        There are a few grammatical and typo errors in the text that need to be re-checked and corrected more carefully.

Reply: We have tried our best to remove typos carefully.

In conclusion, I hope that they find my comments constructive and I am eagerly looking forward to re-review their carefully revised and improved version.

Reply: Thanks for providing comments on our manuscript. We are ready to submit revised manuscript.

Reviewer 3 Report

1. The Table 1 and Table 2 should be removed into ESI.

2. “IR spectra of ligand shows a sharp band at 1628 cm –1 (in I) and 1594 cm –1 (in II) due 196 to (C=N) stretch.” This part should be cited some refs, such as Inorganics, 10(2022) 202 and Micropor. Mesopor. Mat, 341(2022) 112098.

3. I think the authors should explain the mechanism in detail.

4. “Nowadays, instead of using conventional catalysts such as Lewis or Brønsted acid and bases, the new generation of catalysts such as supported catalysts, polymer-based catalysts, metal organic frameworks and enzymes have been used to transform various organic and inorganic transformations.” Some related refs could be cited, such as Org. Chem. Front., 2020,7, 3515-3520; New J. Chem., 2020, 44, 16265-16268; J. Org. Chem. 2019, 84, 14627−14635 and Org. Chem. Front., 2021, 8, 4554–4559

5. Please provide the PXRD for all the complexes.

6. One of the most important factors affecting in photocatalysis is air humidity, etc. Why did the author consider the effect of pH?

7. Source and purity of all chemicals used should be specified in the experimental section.

8. The structural stability should be confirmed by the morphology of samples after multiple tests.

Author Response

Comments and Suggestions for Authors

1. The Table 1 and Table 2 should be removed into ESI.

Reply: As suggested, Tables 1 and 2 have been shifted to supporting materials.

2. “IR spectra of ligand shows a sharp band at 1628 cm –1 (in I) and 1594 cm –1 (in II) due to n(C=N) stretch.” This part should be cited some refs, such as Inorganics, 10(2022) 202 and Micropor. Mesopor. Mat, 341(2022) 112098.

Reply: Unfortunately, paper “Inorganics, 10(2022) 202” does not mention interpretation of n(C=N) stretch, though IR spectra presented there have such IR band. Similarly in other paper, ligand, 6-(4-carboxylphenyl)nicotinic acid has been used to prepare complex where carboxylate group is present for coordination. Band due to ring nitrogen is present but it has not been assigned in either text in main paper or spectrum in SI. However, we have provided a suitable reference for the assignment for azomethine n(C=N) stretch.

3. I think the authors should explain the mechanism in detail.

Reply: Mechanism has been explained.

4. “Nowadays, instead of using conventional catalysts such as Lewis or Brønsted acid and bases, the new generation of catalysts such as supported catalysts, polymer-based catalysts, metal organic frameworks and enzymes have been used to transform various organic and inorganic transformations.” Some related refs could be cited, such as Org. Chem. Front., 2020,7, 3515-3520; New J. Chem., 2020, 44, 16265-16268; J. Org. Chem. 2019, 84, 14627−14635 and Org. Chem. Front., 2021, 8, 4554–4559

Reply: We have cited these references in introduction section.

5. Please provide the PXRD for all the complexes.

Reply: We have recorded P-XRD of homogeneous complexes, their discussion is presented in the text.

6. One of the most important factors affecting in photocatalysis is air humidity, etc. Why did the author consider the effect of pH?

Reply: It seems that reviewer has some confusion here. We have not considered the effect of pH anywhere. Further, catalytic reaction studied here is not photocatalytic.

7. Source and purity of all chemicals used should be specified in the experimental section.

Reply: Details of chemicals purity etc. have been added in the experimental section.

8. The structural stability should be confirmed by the morphology of samples after multiple tests.

Reply: Due to limited assess of instrument, it has not been possible to record morphology of heterogeneous catalyst after multiple catalytic test. However, morphology obtained and elemental mapping recorded strongly suggest the stability of complexes over the solid support.

Round 2

Reviewer 3 Report

accept